# Lightweight Knowledge Graph Construction and Embedded Reasoning for Node-RED Applications

Mirco Soderi[1]

[1]*Mines Saint-Étienne, F-42023, Saint-Étienne, France*

## Abstract

Node-RED is a widely adopted low-code platform for flow-based event-driven applications. Although its community continuously produces reusable components and troubleshooting knowledge (e.g., GitHub issues, forums, flow libraries), this knowledge is not readily available to developers during application design. The same can be said for organisational policies or experiential knowledge. This work presents a lightweight approach to representing both Node-RED user applications and heterogeneous external knowledge as knowledge graphs, enabling different categories of rule-based reasoning to provide actionable insights to developers during development. Example use cases include flagging nodes associated with known issues, detecting reusable community flows, identifying code smells, and checking compliance with organisational policies. To support practical adoption and energy efficiency at scale, a lightweight RDF store and reasoning engine were embedded directly into the Node-RED runtime via a plugin. Reasoning runs on deploy, and its results are stored, and then surfaced within the Node-RED Web editor through a custom sidebar plugin. This required (i) designing a lightweight ontology to model Node-RED user applications, (ii) defining efficient JSON-to-JSON-LD mappings for both application and external data, and (iii) minimising memory and computational overhead through wise tools selection, and resource-aware system engineering. The experimental results confirmed that RDF representations and rule-based reasoning can be integrated into the Node-RED development environment and distributed as a Docker image while keeping image size and runtime resource consumption close to a vanilla Node-RED setup. These results suggest that knowledge graph construction and reasoning can be embedded into development environments with limited overhead, enabling resource-efficient developer-centred semantic assistance during application design.

## Keywords

Knowledge Graph Construction, Semantic Integration, Rule-Based Reasoning, Embedded Reasoning, Low-Code Programming, Event-Driven Programming, Node-RED, Resource-Aware Systems, Design-Time Support

## 1. Introduction

Visual programming environments for flow-based event-driven software development such as Node-RED have significantly lowered the barrier to designing event-driven and IoT applications by allowing users to construct workflows through interconnected functional nodes. Although this accessibility enables rapid development and broader adoption, it also shifts complexity from programming syntax to design decisions that rely on scattered knowledge about nodes behaviour, compatibility, and best practices. This knowledge often remains external to the development environment, forcing developers to rely on documentation, personal experience, or trial-and-error exploration.

This gap manifests itself in several ways. Developers may unknowingly introduce defective or inefficient configurations, overlook reusable components, or construct flows whose structural limitations only become apparent at runtime. Although the required knowledge frequently exists—within documentation, communities, or prior projects—it is not available at the time decisions are made. Consequently, development becomes dependent on fragmented information sources rather than integrated guidance, increasing cognitive load, and reducing reliability as system complexity increases.

The consequences of this knowledge gap extend beyond individual development to organisational contexts, where flows must comply with internal standards, ensure maintainability, and support long-term evolution. In such settings, the absence of contextualised guidance increases the dependency on expert intervention and informal practices, and thus the probability of defective software being created.

*KGCW'26: 7th International Workshop on Knowledge Graph Construction, May 10, 2026, Dubrovnik, Croatia*

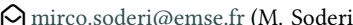 mirco.soderi@emse.fr (M. Soderi)

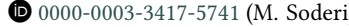 0000-0003-3417-5741 (M. Soderi)

Existing approaches attempt to mitigate these issues through documentation platforms, shared repositories, static analysis tools, or runtime monitoring solutions. Although valuable, these approaches typically remain external to the development workflow or operate after design decisions have already been made. As a result, they provide retrospective insight rather than decision-time support, leaving developers without integrated mechanisms to access relevant knowledge during flow construction.

The central issue is therefore not the absence of knowledge, but its misalignment with the moment of decision-making. Knowledge relevant to flow design—regarding node behaviour, structural patterns, compatibility constraints, or organisational policies—exists, but remains detached from the modelling environment where design choices occur. Addressing this misalignment requires mechanisms that embed contextual knowledge directly into the development process, transforming external information into actionable guidance in design-time.

To this end, this paper proposes a configurable and explainable semantic framework for supporting Node-RED development. The framework integrates structured knowledge representations with rule-based reasoning to enable context-aware analysis and guidance during flow construction. Its design emphasises modularity, scalability, and transparency. The main contributions of this work are:

1. A lightweight ontology to model Node-RED flows, nodes, and selected contextual properties.
2. Methods for representing Node-RED applications and external knowledge as RDF resources.
3. A rule-based reasoning mechanism enabling configurable evaluation of design-time conditions.
4. A modular architecture supporting integration with existing development workflows.
5. A proof-of-concept and experimental validation that demonstrate the feasibility of the approach.

Together, these elements establish a foundation for embedding semantic decision support within the Node-RED platform, with potential applicability to other visual and flow-based development environments, reducing cognitive burden while preserving flexibility.

In the remainder of this paper: Section 2 places the work within existing research; Sections 3–6 describe the modelling strategy, architecture, and integration of semantic reasoning; Section 7 presents the results of the experimental evaluation, followed by a discussion of limitations and future directions.

## 2. State of the Art

The research relevant to this work spans the following five partially overlapped areas:

- Construction of knowledge graphs and semantic integration;
- Ontology-based representation and analysis of software artefacts;
- Use of experiential knowledge from software engineering communities;
- Flow-based, event-driven, low-code development environments;
- Sustainable and energy-sensitive software engineering.

Each component contributes mechanisms for structuring knowledge, supporting development, or assessing sustainability, but, to the best of current knowledge, no work combines these approaches to make heterogeneous knowledge *actionable at design time* within the developer's modelling context. This section positions the present work at the intersection of these lines and clarifies the gap addressed.

### 2.1. Construction of Knowledge Graphs and Semantic Integration

Knowledge Graph Construction (KGC) studies how heterogeneous data sources can be transformed into structured, interoperable representations that support querying and reasoning. Mature mapping languages and construction frameworks enable systematic conversion from heterogeneous sources into RDF graphs, supporting reproducible integration pipelines and reuse across systems [1] [2]. Beyond construction pipelines, surveys and reference works consolidate best practices, quality concerns, and methodological challenges in the building and maintenance of knowledge graphs [3]. Although these approaches provide robust foundations for integrating external information, they typically focus on building and integrating standalone knowledge datasets rather than operationalising such knowledge within an interactive software development workflow under resource constraints.

## 2.2. Ontology-Based Representation and Analysis of Software Artefacts

This line of research investigates ontology-driven representations of programs to support analysis and interoperability. Zhao et al. [4] proposed ontology-based program analysis as a way to standardise program representations and allow multiple analyses to operate on a shared conceptual model, reducing tool-specific preprocessing and fostering reuse. Subsequent work expanded the use of ontologies for software artefacts, including richer models for code structure and semantics [5] and approaches that connect software representations to external knowledge to support analysis tasks [6]. Related efforts also explore ontology-based approaches to support software debugging and analysis, including frameworks that encode program structure and execution traces as knowledge graphs and enable reasoning over software behaviour [7]. In general, these works demonstrate that semantic representations can support software analysis and integration. However, they are typically applied after the software artefact has been created, for verification or debugging purposes, rather than to support developers during the construction of the application. In addition, most approaches focus on representing the software artefact itself and rarely incorporate external knowledge.

## 2.3. Use of Experiential Knowledge from Software Engineering Communities

Software engineering research mines repositories and developer communications to extract experiential knowledge about bugs, practices, and recurring development challenges. Studies in text mining and repository analysis show that issue trackers, commits, and discussions contain signals that can be structured into actionable insights [8]. Systematic reviews on learning from bug reports further highlight the richness of developer-generated artefacts and the potential of automated methods to classify, predict, or prioritise defects [9]. However, these techniques typically externalise knowledge into reports, dashboards, or machine learning models, and rarely output semantic representations that enable automated reasoning and provisioning of integrated contextual support *within* the development environment during design-time decisions.

## 2.4. Flow-Based, Event-Driven, Low-Code Development Environments

Flow-based and low-code environments are widely adopted for event-driven and IoT applications because they enable rapid composition through visual node-based workflows. Their explicit representation of the application structure makes the behaviour of the system appear easier to inspect and modify. Previous work demonstrates usage in very diverse domains, including smart transportation, logistics, energy and manufacturing [10] [11] [12] [13]. Although these ecosystems provide mature tooling for runtime execution and deployment, they offer limited support for design-time guidance: developers still rely on external documentation, experience, or trial-and-error to assess node behaviour, compatibility constraints, and higher-level modelling choices as flows scale.

## 2.5. Sustainable and Energy-Sensitive Software Engineering

Energy-aware software engineering research provides models for estimating energy consumption as a function of hardware resource utilisation, with CPU activity, memory usage, storage operations, and network communication repeatedly identified as dominant contributors to software energy behaviour [14, 15, 16]. This motivates evaluating the environmental sustainability of the proposed approach primarily in terms of CPU and memory overhead, and provides ground for key architectural choices presented later in this work, such as maintaining an in-memory RDF store to reduce disk activity, and embedding semantic functionality within the Node-RED runtime to reduce network traffic.

## 2.6. Research Gap

Taken together, these research directions provide complementary capabilities: knowledge graph construction supports integration of heterogeneous information, ontology-based approaches enable semantic representation and analysis of software artefacts, repository mining structures experiential

developer knowledge, flow-based environments expose application structure in machine-readable form, and sustainable software engineering provides models to estimate energy consumption and guidelines for resource-aware design. However, these strands remain largely disconnected in practice and do not provide a lightweight framework that (i) represents an application *in construction* alongside heterogeneous external knowledge within a unified semantic framework, and (ii) enables configurable, explainable reasoning that surfaces contextual guidance directly inside the development environment. This work represents a first step towards addressing this gap.

## 3. Design Principles and General Approach

This work treats software applications, in this case Node-RED applications, as structured representations whose design decisions require knowledge beyond what is explicitly encoded in the application itself. In particular, two types of knowledge matter at development time: (i) internal application knowledge (in the case of Node-RED, flows, nodes, wiring topology, configuration), which supports structural reasoning, and (ii) external knowledge from variegated documentation and community artefacts (issues, discussions, components), which captures experience about defects, misuse patterns, and reuse opportunities. The proposed approach represents both within a unified semantic framework, enabling them to be related, queried consistently, and filtered by context to provide integrated design-time support.

### 3.1. Design Principles

Three main requirements drive the design:

1. **Development-time support without runtime disruption.** The approach must preserve the interaction model and resource profile of the original development and execution environment, avoiding heavy instrumentation, and keeping runtime overhead low.
2. **Interoperability across heterogeneous knowledge.** The semantic representation must accommodate application structure and external knowledge within a unified framework, enabling filtering by context, cross-source querying, and linkage or annotation through reasoning.
3. **Explainability and configurability.** Guidance must be grounded in explicit, inspectable logic so that users can understand outcomes, adapt rules to local practices, and rely on the semantic subsystem for real-world decision-making.
4. **Relevant and focused feedback.** Reasoning must filter knowledge according to the current application context and associate insights with the relevant software artefacts, ensuring that developers receive precise and actionable guidance rather than generic or excessive information.

### 3.2. Static Modelling of the Application in Context

The application is represented as a knowledge graph that captures both its internal structure and the minimal context needed to assess the relevance of external knowledge (e.g., the version of the Node-RED platform). Nodes, ports, wires, configuration parameters, and flow boundaries are explicitly represented, allowing structural constraints and patterns to be expressed declaratively. This representation is intentionally development-orientated and, therefore, static. The representation and analysis of the application at runtime falls outside of the scope of this work.

### 3.3. Modelling Strategy and Resource Identifiers

Interoperability is obtained by combining (i) a lightweight ontology that models Node-RED concepts, (ii) standard RDF/OWL elements, and (iii) widely adopted vocabularies such as schema.org. The driving criterion is to ensure that the application components and the external knowledge can be linked to each other as appropriate. Resource IRIs are defined in such a way as to establish an immediate correspondence between RDF resources and associated application components or external knowledge.

### 3.4. Rules as Data and Human-Centred Reasoning

Reasoning is implemented as an explicit configurable layer expressed through rules stored and managed as data. This enables (i) transparent explanations linked to the rules that triggered a conclusion, (ii) adaptation to local constraints by adding, removing, or customising rules, and (iii) iterative refinement as developers gain insight into the behaviour of the system and start to play an active role in it by adding their own rules. The final intent is to enable the provisioning of relevant contextual knowledge at development time and to make the guidance auditable.

### 3.5. Lightweight-by-Design: Selectivity and Compression

Resource efficiency is treated as a first-order goal. The approach therefore follows two complementary strategies: **selectivity**, which means ingesting and representing only what is needed for filtering by context and reasoning, and **compression**, which means reducing storage size via deterministic compression of data at rest. Together, these strategies enable development-time support that remains feasible on modest hardware and compatible with the lightweight nature of Node-RED deployments.

## 4. Modelling of Applications and External Knowledge

The approach requires representing both the application and heterogeneous external knowledge within a shared semantic framework while remaining compatible with interactive development environments. The modelling strategy therefore balances interoperability across diverse sources with lightweight representation, focusing on the aspects required for context-based filtering and knowledge linkage and annotation through reasoning, and prioritising relevance over completeness.

### 4.1. Modelling Scope and Objectives

Providing relevant insights to developers at design time requires relating three types of knowledge: (i) application structure, providing a static representation of the application; (ii) execution context, capturing essential characteristics of the technological environment; (iii) external knowledge, representing experience, recommendations, and reusable software available outside the application.

The objective is to enable the contextualised application to be related to relevant external knowledge. This corresponds to being able to answer two categories of questions: (i) contextual filtering questions, e.g., "Is this external document making reference to a specific Node-RED version? If so, does that version match the version of the Node-RED runtime that runs the application?"; (ii) subordinately, linkage or annotation questions, e.g., "Does the type of this specific node that is part of the implementation of the application appear in the title of this external document?" or "Is this specific node part of a loop?".

### 4.2. Ontology Overview

A lightweight ontology (Fig. 1) is proposed to represent the structural aspects of Node-RED applications together with the minimal execution context required for context-based knowledge filtering. The Noy and McGuinness guidelines for ontologies engineering [17] were used as a reference.

#### 4.2.1. Modelling of Node-RED Applications

Node-RED applications expose a graph structure composed of interconnected nodes organised into flows and subflows. The model preserves this structure by representing flows as program units and nodes as executable components linked through explicit wiring relations.

Node types are modelled as literal properties rather than ontology classes, reflecting their role as identifiers originating from external package repositories rather than stable conceptual categories. This avoids unnecessary ontology growth and allows new node types to be represented as needed without requiring any ontology maintenance effort.

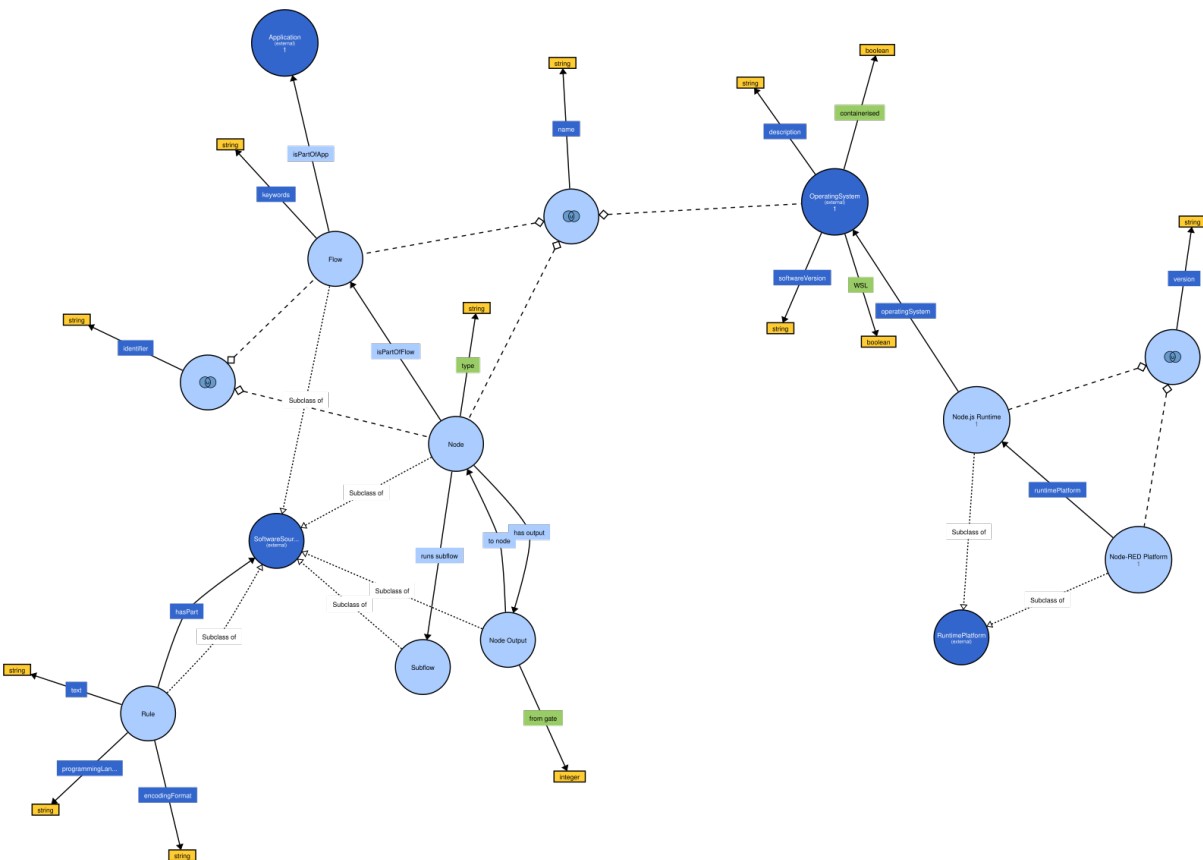

**Figure 1:** High-level overview of the proposed ontology. The newly introduced classes and predicates are represented along with the terms reused from schema.org, either directly or as umbrella terms.

**Node-RED (simplified JSON)**

```
tab: id=f0ac...867a, label="Example"

inject: id=d435...54ce,
        type="inject", name="Event",
        wires=[[7731...8d7c]]

debug: id=7731...8d7c,
        type="debug", name="Log",
        complete="payload"
```

Example

⬜ ➔ Event — Log ☰ ⬜

**RDF (key triples)**

```
urn:nrua:f0ac...867a a nrua:Flow ;
  schema:name "Example" .

urn:nrua:d435...54ce a nrua:Node ;
  nrua:type "inject" ; schema:name "Event" ;
  nrua:isPartOfFlow urn:nrua:f0ac...867a ;
  nrua:hasOutput urn:nrua:od435...54ce0 .

urn:nrua:od435...54ce0 a nrua:NodeOutput ;
  nrua:fromGate 0 ;
  nrua:toNode urn:nrua:7731...8d7c .

urn:nrua:7731...8d7c a nrua:Node ;
  nrua:type "debug" ; schema:name "Log" ;
  schema:additionalProperty id=urn...complete .

urn...complete a schema:PropertyValue ;
  schema:name "complete"; schema:value "payload" .
```

**Figure 2:** Simplified correspondence between a basic Node-RED flow and its RDF representation.

The configuration parameters that are specific of specific node types are represented as generic additional properties attached to node instances rather than dedicated ontology predicates. Since these parameters are defined by externally maintained packages and evolve independently of the ontology, treating them as ontology predicates would introduce a disproportionate maintenance effort while providing a limited reasoning benefit.

Connections between nodes are explicitly modelled to support structural reasoning tasks such as reachability analysis, detection of unused processing paths, and identification of cyclic message flows. Figure 2 illustrates a simplified correspondence between a Node-RED flow and its RDF representation.

**GitHub Issue (simplified JSON)**

```
issue: number=5449,
    html_url="https://github.com/.../issues/5449",
    title="Error when using a subflow ..."
    created_at="2026-01-23T15:10:34Z",
    labels=["needs-triage"],
    score=6

... many other GitHub fields omitted ...
```

**RDF (key triples)**

```
<https://github.com/.../issues/5449>
  a schema:DigitalDocument ;
  schema:title "Error when using a subflow ..." ;
  schema:date "2026-01-23T15:10:34Z" ;
  schema:url <https://github.com/.../issues/5449> ;
  schema:category urn:term:issues:label:needs-triage ;
  schema:contentRating urn:rating:issues:5449 .
urn:term:issues:label:needs-triage
  a schema:DefinedTerm ; schema:name "needs-triage" .\
      cite{noy2001ontology}
urn:rating:issues:5449
  a schema:Rating ; schema:ratingValue 6 .
```

**Figure 3:** Simplified mapping example of a GitHub issue to its RDF representation.

### 4.2.2. Modelling of Execution Contexts

Community discussions and issue reports often reference specific operating systems, runtime versions, or platform releases. To enable context-based filtering, minimal selective modelling of the technology stack is introduced. Only attributes that commonly appear in external knowledge are represented. This selectivity allows for low-resource relevance assessment.

### 4.3. Modelling of External Knowledge

External knowledge is represented as first-class graph data rather than accessed through external queries on-demand at runtime. This enables efficiency, reliability, selectivity, and sustainability while supporting direct linking between application elements and heterogeneous pieces of external knowledge, which can take the form of textual content or community-provided software artefacts.

Textual knowledge such as community discussions, issue reports, and documentation entries is modelled as digital documents enriched with essential metadata (title, date of publication, Web address) and possible contextual information (operating system, Node.js version, Node-RED version), if explicitly mentioned. This reflects the purpose of the representation, which is uniquely to enable context-based filtering, linkage to software components, and basic discoverability, and allows for resource-efficient ingestion and storage while allowing documents to be linked to application components through simple signals, such as mentions of specific node types or platform versions that appear in document titles. Figure 3 illustrates a simplified mapping of a GitHub issue to its RDF representation.

Community-provided flows and configurations are represented through minimal descriptive metadata and summarised structural characteristics, such as the set of node types involved. Full structural import is intentionally avoided, as it is unnecessary for current reasoning tasks and would increase storage and ingestion costs without proportional benefit.

## 5. Architecture and Implementation

The proposed extension of the Node-RED platform enhances the original platform rather than replacing it, integrating knowledge representation and reasoning capabilities while preserving compatibility, usability, and the lightweight characteristics of the vanilla Node-RED runtime.

The architecture (Fig. 4) comprises five main components: (i) an extended Node-RED runtime that provides semantic capabilities along with standard execution; (ii) an embedded knowledge store (μRDF) that maintains RDF representations of applications, context, external knowledge, rules, and inference; (iii) a reasoning layer that executes configurable rules and persists results in the knowledge store; (iv) an external knowledge ingestion pipeline (not depicted); (v) a sidebar panel integrated in the native Node-RED Web editor for user-friendly inspection and management of semantic data.

Semantic functionality is introduced through modular extensions aligned with the native Node-RED architecture. At startup and on application update (deployment), the native flows representation (JSON)

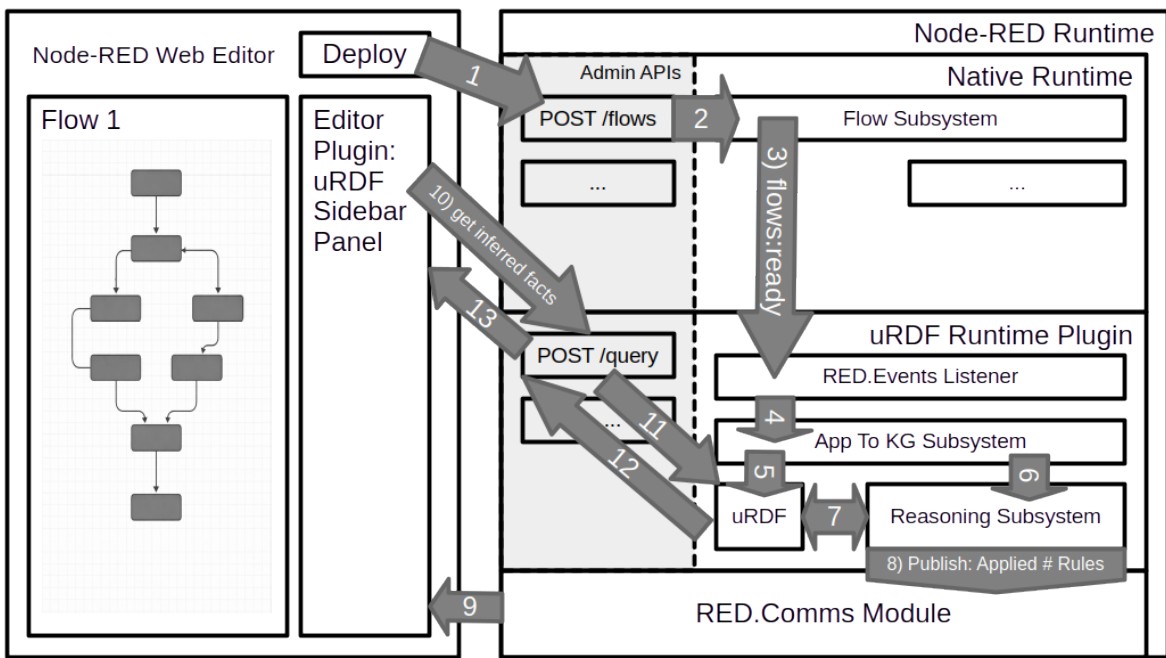

**Figure 4:** High-level architecture showing the sequence of interactions during app deployment.

is converted into RDF and stored in the embedded knowledge store, and then reasoning is operated and its results are stored in the embedded knowledge store. Instead, external knowledge acquisition is decoupled from application deployment and is handled by independent importers that retrieve and transform heterogeneous sources before loading them into the knowledge store via Admin APIs.

## 5.1. Integration with the Node-RED Runtime

Semantic capabilities are integrated using the standard Node-RED extension mechanism. Flows continue to be executed by the unmodified engine, while semantic services operate alongside existing runtime functionality. This approach preserves portability, minimises coupling with internal platform details, and allows independent evolution of the semantic layer.

**Semantic updates are linked to the standard deployment lifecycle.** Upon deployment, which corresponds to the act of saving changes made in the user application, the new implementation is retrieved, and its native JSON representation is programmatically transformed into a JSON-LD representation that conforms to the lightweight ontology presented in Section 4. Declarative mapping frameworks such as RML and related approaches [1, 2], as well as iterator-based solutions including SPARQL-Generate [18], were evaluated for graph construction, as alternatives to the programmatic approach, but could not express the required mapping without preprocessing or engine extensions due to the peculiar structure of the JSON documents that are natively used to represent Node-RED flows.

**Semantic data administrative operations are exposed through standard Node-RED extension interfaces.** This allows data inspection, management, and basic monitoring while remaining within the existing runtime and without introducing additional deployment units. Indeed, the new operations become part of the platform runtime along with native operations, and therefore they also benefit from the platform security policy and implementation, with no development effort required to this end.

**At startup, the runtime is initialised with predefined semantic assets.** These include the ontology and the reasoning rules. The predefined reasoning rules are to be considered as a proof-of-

concept and include flagging nodes associated with known issues, detecting reusable community flows, identifying code smells, and checking compliance with hypothetical organisational policies.

## 5.2. Knowledge Store Selection and Optimisation

The programmatic construction of the application knowledge graph, tightly coupled to the deployment lifecycle, requires a storage design that prioritises predictable behaviour, lightweight integration, and minimal operational overhead. Rather than deploying an external triple store, the architecture adopts a lightweight in-memory RDF store integrated directly into the Node-RED runtime and resulting from the customisation of the μRDF store [19].

**Controlled operational model.**   Since all data uploaded to the RDF store are generated in a controlled environment, assumptions can be made that would not be possible to make in an open, generic environment. Such assumptions allowed to disable specific features of the RDF store and proceed to peculiar optimisations, which allowed to achieve significant improvements in baseline resource efficiency, while preserving the level of functionality that is required to support the specific reasoning to be performed and the specific functionalities to be offered to developers. For example, JSON-LD documents are flattened prior to storage using a custom transformation implementing only the subset of processing required for our specific purposes, avoiding the overhead of full general-purpose JSON-LD processing. Additionally, frequently occurring vocabulary terms are internally compressed through a lightweight IRI substitution mechanism applied transparently during data insertion, query execution, and result retrieval. To perform IRI substitution without having to rely on heavy parsing, support was removed for specific constructs in JSON-LD documents and SPARQL queries. Beyond that, the native μRDF store itself, designed for low-resource settings, supports a limited SPARQL dialect.

## 5.3. Reasoning Layer Integration

Reasoning is implemented as part of the runtime extension and is triggered as a finite task during application deployment, after reconstruction of the application knowledge graph, ensuring that reasoning operates on a stable and fully normalised representation of the deployed application while keeping resource overhead bounded. Different rule types are supported, and results are stored in the RDF store.

**Deployment-coupled reasoning model.**   Reasoning is executed as a bounded runtime task aligned with the deployment lifecycle. During deployment, the runtime regenerates the application knowledge graph and then invokes inference, producing derived knowledge that reflects the current configuration. This avoids persistent reasoning processes while keeping execution predictable and overhead bounded.

**Rule representation and execution.**   Inference rules are stored as first-class resources within the knowledge graph rather than embedded in the runtime implementation or loaded as a startup configuration. The set of rules is retrieved from the RDF store at the beginning of each reasoning execution, allowing the reasoning configuration to evolve independently of the runtime extension.

**Two complementary rule modalities are supported.**   *SPARQL rules* express pattern-based derivations executed directly against the store, with query bindings converted into inferred triples. *N3 rules* are executed using the Eyeling reasoner [20], a lightweight JavaScript system related to the EYE reasoner [21]. Each N3 rule specifies an associated SPARQL projection used to extract a bounded set of relevant facts, which are provided to the reasoner together with the rule program. Intermediate predicates produced during reasoning are treated as internal artefacts and are not exposed to Node-RED developers.

**Inference lifecycle and storage.**   Inference follows a deterministic rebuild model. Derived triples are computed as a finite operation and written to a dedicated named graph that atomically replaces previously inferred results. This streamlines the update of inferred knowledge and guarantees that

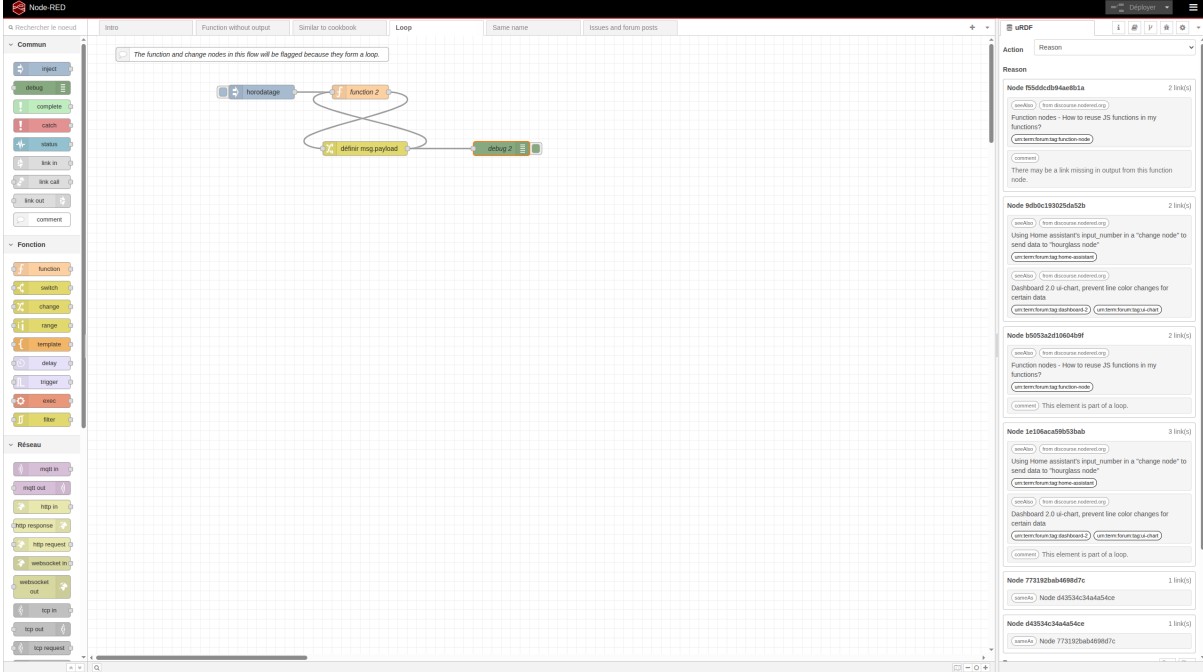

**Figure 5:** Node-RED Web editor with example flows and inference results shown in the new uRDF sidebar panel.

inferred knowledge depends solely on the current deployment and rule set, preventing residual state or ordering effects. The inferred graph remains subject to the same controlled storage assumptions as the asserted data, preserving consistency with the lightweight runtime architecture.

## 5.4. External Knowledge Integration

External knowledge integration extends the knowledge base beyond the application structure by incorporating different types of external knowledge (e.g., community, organisation, or experiential knowledge) while preserving the controlled data ingestion and execution model established in this section. Indeed, external data are introduced through dedicated import tools that transform documents into representations compatible with the storage and processing constraints of the extended platform.

**Controlled integration.** Imported knowledge conforms to the same assumptions applied to internally generated data, including flattened JSON-LD representation, reduced JSON-LD expressivity, and compression at rest. Being imported knowledge, part of the normalisation effort is offloaded to the import tools, thus reducing the overhead on the Node-RED runtime.

**Different categories of knowledge are imported for proof-of-concept purposes.** These include GitHub issues from the Node-RED GitHub repository, forum posts from the Node-RED Discourse forum, and software artefacts (flows) from a GitHub repository. They are first downloaded via Web interfaces or API requests, and then made available to the import tools as locally-available artefacts.

## 5.5. Node-RED Web Editor Extension: the Human-Computer Interaction Layer

Semantic capabilities are exposed directly within the standard Node-RED Web editor through a dedicated sidebar panel, named *uRDF* (Fig. 5), implemented as a Node-RED editor plugin. The interaction layer provides controlled access to graph construction, reasoning results, rule management, and knowledge exploration within the existing development workflow.

**Inference inspection.**   The primary interaction is inspection of inferred knowledge through a *Reason* action, which presents the results of deployment-triggered inference in a user-friendly form. Developers therefore observe derived knowledge directly within the familiar editor environment, without interacting with RDF syntax or external tools.

**Rule configuration.**   Inference rules, internally stored as RDF resources, are managed through dynamically generated form-based interfaces consistent with standard Node-RED configuration dialogs. On submission, the updated rules are transformed into JSON-LD documents and stored in a dedicated graph in the RDF store, allowing reasoning behaviour to be modified while leaving the runtime extension unaltered. Subsequent application deployments automatically use the updated rule definitions.

**Knowledge exploration.**   The sidebar supports the inspection of stored knowledge. A *Search* action enables retrieval of resource or graph descriptions formatted as JSON-LD documents, while a *Query* action exposes (not so) arbitrary SPARQL execution with results displayed in the same interface. An *Export* action allows complete graphs to be downloaded as JSON-LD documents for external reuse.

**Developer's habits are respected.**   By integrating inference inspection, rule configuration, and knowledge loading and exploration into the existing Node-RED Web editor that developers use daily to create their applications, semantic functionality augments established Node-RED practices without altering the core development workflow, so respecting the developer's habits.

## 6. Experimental Evaluation

The evaluation assesses whether semantic representation and deployment-coupled reasoning can be integrated into Node-RED while preserving lightweight runtime characteristics. The experiments compare a vanilla Node-RED platform with the enriched Node-RED platform presented in this work.

### 6.1. Image Size Overhead

The enhanced image introduces a very small storage overhead compared to the vanilla Node-RED image, quantifiable in about 3%, with negligible implications for storage consumption and network transfer when distributing or updating the container image.

### 6.2. Baseline Overhead

Embedding the semantic extension increased baseline memory usage by approximately 20 MB of RAM at rest, corresponding to nearly +50% compared to vanilla Node-RED. CPU utilisation remained negligible in both configurations. The primary impact was therefore structural memory overhead.

The isolated loading of the uRDF module in a minimal Node.js application showed an increase of about 22 MB of RAM, confirming that the RDF store accounted for most of the baseline overhead.

Further decomposition revealed that approximately 17 MB were attributable to general-purpose JSON-LD processing dependencies and about 4 MB to auxiliary input/output mechanisms. Since the platform operates under controlled document assumptions, these capabilities were unnecessary.

Introducing a stricter contract for the uRDF store and in particular relocating JSON-LD flattening outside of it and removing support for data load from URL reduced the idle memory footprint from about 20 MB to about 3 MB of RAM. CPU utilisation remained negligible.

### 6.3. Scalability of the Semantic Representation of the Node-RED User Application

Scalability was evaluated using progressively larger Node-RED applications artificially generated by wisely replicating a small set of flows. Memory usage scaled gracefully, increasing predictably with application size due to growth of the application knowledge graph, while CPU utilisation remained low.

However, for large applications, the gap between enhanced and vanilla Node-RED peaked at +50 MB of RAM. To reduce the memory occupation of the knowledge graph that represents the Node-RED user application, the IRI compression mechanism was introduced, which reduced memory consumption by approximately 50%, while slightly increasing CPU usage, which peaked at 10% on a single core configuration.

These results indicate that a significant portion of the scaling overhead originates from representation redundancy. In controlled environments, lightweight deterministic compression can therefore provide efficiency gains without altering the exposed semantic model.

### 6.4. Scalability of Reasoning

With reasoning enabled, memory consumption continued to scale primarily with the size of the application knowledge graph. The additional overhead introduced by rule execution remained bounded and proportional to the complexity of the Node-RED user application being analysed.

CPU utilisation increased during redeployment events, but returned to idle levels thereafter. The end-to-end redeployment time remained below 200 ms even for very large applications.

Overall, the results demonstrate that semantic representation and rule-based inference can coexist within a Node-RED runtime while maintaining bounded and predictable resource behaviour, provided that storage and processing assumptions are aligned with the operational context and that subsequent optimisations are introduced.

## 7. Discussion

### 7.1. Lessons Learned

The results of Section 6 show that semantic functionality can remain lightweight when designed to fit the structure of the host development environment. The main efficiency gains did not come from optimising reasoning algorithms, but from architectural choices: relying on lightweight dependencies, building the application graph programmatically at deployment time, running reasoning as a bounded task, and tightly controlling how data are loaded, stored, and accessed.

The experiments also indicate that the observed resource overhead was mainly due to general-purpose RDF features intended for generic data processing. When data are produced within a controlled and predictable context, processing mechanisms can be tailored to that context, and in such a way, resource utilisation, and in particular memory usage, can be reduced without sacrificing system functionality.

A further lesson concerns determinism. Rebuilding both the application graph and the inferred graph at each redeployment prevents state accumulation and keeps system behaviour predictable. This rebuild strategy supports lightweight operation by simplifying the implementation and improving the efficiency of the reasoning subsystem. Overall, the findings suggest that efficient semantic integration depends more on architectural coherence than on isolated performance optimisations.

### 7.2. Contextualisation of Results

The approach presented in this work operates under explicit assumptions that delimit its applicability. The optimisation strategies rely on controlled graph generation and querying, and therefore do not apply to open-world scenarios in which such a level of control cannot be achieved. Similarly, the strict operational model and, in particular, the restricted SPARQL dialect constrain query flexibility in order to maintain bounded resource usage, but that is only feasible in scenarios in which control over the submitted queries can be maintained. The experimental results and their associated evaluation should therefore not be interpreted as general performance claims about RDF systems or reasoning engines. Instead, they demonstrate that lightweight semantic integration is feasible in environments where the operation boundaries and data management mechanisms can be clearly defined and effectively enforced. In contexts that require greater flexibility or expressivity, different trade-offs may apply.

### 7.3. The Foreseeable Future for Developer-Centric Design-Time Semantic Assistance

Beyond architectural feasibility, the broader objective of this work is to take a first step toward using Semantic Web technologies to help developers build higher-quality applications more efficiently by providing relevant, contextualised insights within the development environment while software is still being created. The present contribution establishes the design principles and infrastructure foundation upon which such assistance can be systematically developed. Building on this foundation, future research may evolve along several complementary and interrelated paths.

The analysis of community, organisational, and normative knowledge could reveal recurring patterns, requirements, doubts, and failure modes, helping to identify developer needs more systematically. Natural language processing and software requirements engineering techniques may serve as analytical tools. Importantly, these tools would not necessarily need to become part of the development environment but could instead inform future research directions.

Although some mapping approaches were explored in this work, future research may further investigate the adoption of standards to decouple transformation logic from runtime, allowing graph generation and runtime capabilities to evolve independently and enabling integration of additional knowledge sources through declarative mappings rather than custom import tooling, improving portability. Research on *lightweight* mapping engines may complement this direction.

Future work may also examine whether richer semantic representations could unlock additional support capabilities. In Node-RED, this may involve automated generation of semantic descriptions for the different types of node, which could improve discoverability and automated service composition, as explored in recent research on industrial IoT [22]. Lightweight implementations of Semantic Web technologies may complement this direction.

Human–computer interaction research may also play a key role. The design and evaluation of developer-facing semantic feedback should be co-developed with expertise in cognitive load, information presentation, and workflow integration. Semantic intelligence is more valuable if presented in ways that improve comprehension and reduce development effort. Consequently, assistance mechanisms should be evaluated in terms of developer performance and understanding.

Finally, future research may explore adapting the methodological approach presented in this work to flow-based and event-driven platforms beyond Node-RED. Over longer horizons, similar principles could inform semantic assistance across other programming paradigms and development environments, even though developer needs and integration patterns are expected to vary significantly across contexts.

## 8. Conclusion

This work investigated the extent to which Semantic Web technologies can be integrated into a lightweight development environment for flow-based event-driven applications without compromising responsiveness or resource profile. Although semantic technologies provide powerful mechanisms for structuring and reasoning over software-related knowledge, their integration within everyday development environments has been limited by complexity and resource requirements.

For the specific case of Node-RED, the results show that semantic capabilities can be embedded directly within the platform runtime through architectural alignment rather than algorithmic optimisation. By coupling graph construction and reasoning to deployment events, constraining data ingestion and querying, and introducing controlled storage and representation mechanisms, the system achieves bounded and predictable resource behaviour while preserving sufficient rule-based inference capabilities and interoperability with semantic standards. The evaluation indicates that the main challenges arise from mismatches between general-purpose flexibility and operational context, and that carefully designed constraints can enable practical lightweight deployment.

Beyond feasibility, this work establishes a foundation for inline, contextualised developer-centred semantic assistance at development time through knowledge integration. The semantic extension exposes application knowledge alongside external knowledge in forms that can be inspected, related, and progressively leveraged to support developers in understanding and improving their systems.

Rather than positioning semantics as an external analytical layer, the approach integrates it directly into the development environment and workflow.

Future research may focus on systematically identifying developer knowledge needs, standardising knowledge graph creation, exploring richer semantic representations, designing resource-efficient tooling, and co-designing interaction mechanisms that translate semantic reasoning into actionable development support. In this perspective, the work represents an initial step toward development environments in which semantic reasoning becomes a native capability, enabling developers to build more reliable applications with lower effort and bounded environmental impact.

## Declaration on Generative AI

During the preparation of this work, the author used Chat-GPT 5.2 to draft contents, and Writefull to improve writing style. After using these tools/services, the author reviewed and edited the content as needed and takes full responsibility for the publication's content.

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

## A. GitHub Repository

The implementation supporting this work is available under a MIT licence at the following address:

https://github.com/mircosoderi/nodered-ontology-based-program-analysis/

The repository contains the complete artefact used to investigate the integration of lightweight semantic technologies within the Node-RED runtime, which enables reproduction of the architectural design and experimental results presented in this paper. In particular, it includes (i) an enriched Node-RED Docker image embedding a lightweight RDF store and N3 reasoner through custom runtime and editor plugins, complete of predefined example flows and reasoning rules for proof-of-concept purposes, (ii) the associated OpenAPI specification, (iii) a lightweight ontology to model Node-RED user applications, (iv) documented and reproducible experiments on resource use and scalability, and (v) JSON-to-JSON-LD converters to import external knowledge.