# OpenReview forum: "Lightweight Knowledge Graph Construction and Embedded Reasoning for Node-RED Applications"
_eswc-conferences.org/ESWC/2026/Workshop/KGCW — Submitted to KGCW 2026_

### Official Review · ~Eduard_Kamburjan1 · 2026-04-01
**Doubts about novelty**

**Rating:** 4
**Confidence:** 4

**Review:**

This paper proposes a framework to serialize Node-RED flows as RDF, and subsequently use this serialization to connect with external knowledge and use this connection to reason over the flow/program.

While the paper is well-written and understandable, I have several doubts about it.

First, there is not much knowledge graph construction, the only graph that is constructed is essentially a program serialization. I do not think that this is the right venue for this work.

Second, it is not novel. Serializing a program or model as a knowledge graph, and a Node-RED flow is a model, to connect with semantic or graph technologies has been explored before, most recently at ICSE (https://dl.acm.org/doi/10.1145/3597503.3623319). It has also been explored by the semantic web community (https://ceur-ws.org/Vol-2980/paper327.pdf, also see 10.4230/TGDK.4.1.3) and in the context of model-driven engineering (https://ieeexplore.ieee.org/document/11273258/), with the latest publication I am aware of being.

Lastly, the introduction states that the main advantage of this approach is traceability of the flow to the requirements and other context documents, by including them in the external knowledge. Beyond my doubts about the novelty of this idea, the paper does not actually substantiate this. The example in section 6 does not demonstrate context modeling and the ontology does not consider it. Section 4.3 is short and seems naive compared to the rich base of knowledge about artifacts and contexts from software engineering, see e.g. https://dl.acm.org/doi/abs/10.1145/3379597.3387442 just for those artifacts occurring in the immediate context of the repository.

I think that this paper would be more suited for a MODELS workshop.

---

### Official Review · ~Samaneh_Jozashoori1 · 2026-04-02
**A system paper: applying knowledge graphs in software engineering**

**Rating:** 6
**Confidence:** 3

**Review:**

Summary of the paper:

The paper proposes a lightweight, embedded semantic framework for the Node-RED low-code event-driven programming platform. The core idea is to represent both a Node-RED user application and heterogeneous external knowledge as RDF knowledge graphs and then to trigger rule-based reasoning at deployment time to surface actionable guidance to developers directly inside the Node-RED web editor.

Importance and relevance of the topic:

The research addresses a well-identified problem: the misalignment between when design decisions are made and when relevant knowledge is made available. Therefore, the main issue this work tackles is not the absence of knowledge, but its misalignment with the moment of decision-making. The contributions of this resource work is an intersection of knowledge graph construction, software engineering support, and sustainable/energy-aware computing which are relevant topics to the Semantic Web community and KGCW.

Main contributions:

The paper claims five contributions: (1) a lightweight ontology for Node-RED flows and nodes; (2) methods for representing Node-RED JSON and external knowledge as RDF; (3) a configurable rule-based reasoning mechanism; (4) a modular architecture embedded in the Node-RED runtime; (5) a proof-of-concept implementation with experimental validation.

Novelty:

As also mentioned in the paper, the general idea of using ontologies and knowledge graphs to support software analysis is well established. What is new is the specific combination of: (a) representing an application under construction together with its external community knowledge in a single unified semantic graph; (b) triggering reasoning at deploy time rather than post-hoc; (c) doing all of this with resource constraints tight enough to coexist with a lightweight IoT runtime; and (d) targeting a flow-based visual programming environment rather than a traditional programming language.

Comment/required improvement: Semantic Node-RED seems to be a close related work and is mentioned briefly in this paper (reference 22) as well, however, the distinction from that work deserves extended discussion in the related work section.

General Comments

* Overall, it is a well-structured and well-written system paper.
* Design principles are well-motivated and explained.
* The experimental results are explained contextualized, honestly, and scientifically responsible, e.g., “The optimisation strategies rely on controlled graph generation and querying, and therefore do not apply to open-world scenarios in which such a level of control cannot be achieved”
* The resource and experiments are publically available.
* The design choices seem to be supported by domain knowledge and not just knowledge of semantic technologies. This increases the reusability of the system.

Detailed comments and required improvements:

* My primary comment that I consider both essential and feasible to address, concerns Section 5, where the architecture presented in Figure 4 is described. (I) Figure 4, which is the main figure of this paper as it describes the architecture, tries to be two things at the same time and does neither well. The caption of the figure says it shows “the sequence of interactions during app deployment.” But the figure simultaneously tries to be a static architecture diagram (showing components and their containment) and a sequence diagram (showing numbered temporal steps). Mixing them into one diagram creates frustration for a reader; following the numbers gets repeatedly interrupted by needing to locate where they are in the containment hierarchy, and vice versa. (II) The text of Section 5, if read independently, is actually clear and well-structured. However, figure 4 which is supposed to illustrate what the text describes is not delivering, and the text does not walk through the figure step by step either. They are parallel rather than mutually conveying the same explanation.

Revising both the figure and the corresponding architectural description to ensure alignment would significantly improve the clarity and overall quality of the paper.

* The evaluation assesses resource consumption but provides no evidence of the system’s core value proposition: whether the inferred insights are actually useful to developers. There are no user studies. From an empirical software engineering perspective, a system can be lightweight and still be useless or misleading. The paper acknowledges this as future work, but I think some pilot evidence of usefulness would significantly strengthen the contribution.

---

### Official Review · ~Xuemin_Duan1 · 2026-04-04
**A system paper with promising motivation but insufficient KGC technical details**

**Rating:** 6
**Confidence:** 2

**Review:**

This paper proposes a framework that embeds KG construction and rule-based reasoning into the Node-RED low-code development environment. The motivation is that the knowledge relevant to Node-RED developers, including known issues, reusable components, and organisational policies, exists but is not available at the moment design decisions are made. The framework represents both the internal structure of Node-RED applications and external heterogeneous knowledge (GitHub issues, forum posts, community flows) as RDF graphs, and triggers rule-based reasoning at deployment time to surface contextual guidance within the editor. I acknowledge the clarity of the problem definition and the solid engineering work with the resource efficiency experiments. However, although I understand the overall approach architecture, the absence of several key technical details makes it hard for me to understand how the system actually works and what level of performance it can be expected to achieve.

**Importance and Relevance**

This paper has a partial fit with the KGC workshop. The central research goal of this paper is to embed KG technology into a development environment to assist developers, with KGC serving more as an implementation means (especially when it skips some KGC details) than as the primary research subject. In this sense, its more central contributions belong to the software engineering and developer tooling communities, but this does not prevent the paper from appearing at this workshop as an in-use or experience contribution, while the paper can better discuss what the specific KGC contribution of this work is.

**Novelty**

The novelty of this paper lies in embedding KGC and rule-based reasoning directly into the runtime of a lightweight low-code platform, unifying the application's internal structure and external community knowledge within a shared semantic framework.

**Major Comments**

- The KGC work details are missing: The KGC of external knowledge, e.g., GitHub issues, receives almost no technical description beyond a reference to "dedicated import tools." This gap is particularly notable for a paper submitted to a KGC workshop, and makes it hard for me to understand what the produced KG looks like. For example, I don't understand whether the content of a GitHub issue is extracted via entity relationship extraction or just transformed as a whole object value.

- The reasoning rules and reasoning results are unclear: The approach mentions reasoning rules, including SPARQL rules and N3 rules, but there is no concrete example rule presented, which makes it difficult to understand how complex the reasoning rules are, whether they go meaningfully beyond metadata-based keyword matching on pure documents, and what the inferred triples look like. I understand that this paper aims to demonstrate feasibility rather than to provide a full evaluation of reasoning quality, and a systematic assessment can reasonably be left as future work. But showing at least one or two concrete rules and a corresponding end-to-end reasoning example is helpful.

- A discussion about how this approach compares to document retrieval might be interesting. Especially when I didn't see the entity relationship extraction work on the GitHub issue content, I was wondering why the work does not just use pure documents for retrieval, and how reasoning on RDF compares to retrieval on documents in terms of expected precision and recall.

- The decision to move away from RML is unclear. The paper mentions "could not express the required mapping without preprocessing or engine extensions due to the peculiar structure of the JSON documents"; a detailed discussion of what specific structural characteristics of JSON made declarative mapping approaches unsuitable is helpful for a KGC workshop.

**Minor Comments**

- Figure 1 is difficult to read since the font size is too small.
- The "Three main requirements drive the design," in section 3.1 seems should be "Four".
- Experiments only focus on resource efficiency, a user study or an experiment to evaluate the usefulness of the approach would be helpful.

This paper addresses a meaningful problem and delivers a well-implemented system with resource efficiency results; however, it misses technical details around the KGC part of the work. It would be helpful if the paper could be revised to clearly explain the KGC details to better align with the KGC workshop.

---

### Decision · Program_Chairs · 2026-04-09

Reject